# Allelic variation of *Escherichia coli* outer membrane protein A: Impact on cell surface properties, stress tolerance and allele distribution

**Chunyu Liao[1], Miguel C. Santoscoy[2], Julia Craft[3], Chiron Anderson[3], Michelle L. Soupir [4], Laura R. Jarboe [1,2]***

**1** Interdepartmental Microbiology Graduate Program, Iowa State University, Ames, Iowa, United States of America, **2** Department of Chemical and Biological Engineering, Iowa State University, Ames, Iowa United States of America, **3** Department of Chemical and Biological Engineering, Biological Materials and Processes (BioMAP) NSF REU Program, Iowa State University, Ames, Iowa, United States of America, **4** Department of Agricultural and Biosystems Engineering, Iowa State University, Ames, Iowa, United States of America

\* ljarboe@iastate.edu

**Data Availability Statement:** All relevant data are within the manuscript and its Supporting information files.

## Abstract

Outer membrane protein A (OmpA) is one of the most abundant outer membrane proteins of Gram-negative bacteria and is known to have patterns of sequence variations at certain amino acids—allelic variation—in *Escherichia coli*. Here we subjected seven exemplar OmpA alleles expressed in a K-12 (MG1655) Δ*ompA* background to further characterization. These alleles were observed to significantly impact cell surface charge (zeta potential), cell surface hydrophobicity, biofilm formation, sensitivity to killing by neutrophil elastase, and specific growth rate at 42˚C and in the presence of acetate, demonstrating that OmpA is an attractive target for engineering cell surface properties and industrial phenotypes. It was also observed that cell surface charge and biofilm formation both significantly correlate with cell surface hydrophobicity, a cell property that is increasingly intriguing for bioproduction. While there was poor alignment between the observed experimental values relative to the known sequence variation, differences in hydrophobicity and biofilm formation did correspond to the identity of residue 203 (N vs T), located within the proposed dimerization domain. The relative abundance of the (I, δ) allele was increased in extraintestinal pathogenic *E. coli* (ExPEC) isolates relative to environmental isolates, with a corresponding decrease in (I, α) alleles in ExPEC relative to environmental isolates. The (I, α) and (I, δ) alleles differ at positions 203 and 251. Variations in distribution were also observed among ExPEC types and phylotypes. Thus, OmpA allelic variation and its influence on OmpA function warrant further investigation.

**Funding:** Funding for this work was provided by the National Science Foundation (https://www.nsf.gov/) grants CBET-1236510 (MLS) and CBET-1604576 (LRJ), and the Unites States Department of Agriculture National Institute of Food and Agriculture (https://www.nifa.usda.gov/), award number 2017-6702-26137 (LRJ). The funders had no role in study design, data collection and analysis, decision to publish, or preparation of the manuscript.

**Competing interests:** The authors have declared that no competing interests exist.

# Introduction

The outer surface of the microbial membrane plays a crucial role in interaction of the cell with its environment through, for example, attachment to biotic and abiotic surfaces, and through transport of chemical species into and out of the cell. Microbial attachment to surfaces is known to be impacted by variation in cell surface properties, such as hydrophobicity and the presence and abundance of various proteins and sugars [1]. Some of these same cell surface properties have also been implicated in microbial tolerance to harsh growth conditions [2–4]. It is desirable to be able to predictably engineer the properties of the microbial cell membrane for biotechnology applications [2,5–7]. For example, genetic modification to alter cell surface hydrophobicity, such as by alteration of cell surface lipopolysaccharide and proteins, has been shown to impact microbial robustness and production performance [3,8–11]. Here, we investigate the effect of naturally-occurring allelic variation in Outer Membrane Protein A (OmpA) on the cell surface properties of *E. coli* and further explore the distribution of these alleles among microbial isolates.

OmpA is a highly abundant and highly characterized protein embedded in the outer membrane of Gram-negative bacteria and is widespread among Gram-negative bacteria, as described elsewhere in many review articles, such as [12–17]. OmpA has been characterized as a porin for nonspecific diffusion of small molecules [18], a target for host immune system defense [19–22], receptor of colicin and bacteriophage [23–27], mediator of plasmid conjugation [28], evasin for pathogenesis of neonatal meningitis-causing *E. coli* (NMEC) and adhesin for attachment to both biotic and abiotic surfaces [1,29–32]. It also acts as a rivet tethering the outer membrane to the cytoplasmic membrane, contributing to the maintenance of cell shape and the integrity of the cell envelope [33,34].

OmpA is a 325 amino-acid transmembrane peptide with a 21 amino-acid signal peptide [17,35–37] and a flexible linker that connects the C-terminal domain and N-terminal domains. Three distinct OmpA conformations have been described–monomeric, dimeric, and 'large pore'. In each of the three conformations, the 171-residue N-terminal domain translocates across the outer membrane eight times, forming eight β-sheets imbedded in the lipid bilayer, four loops on the cell surface and four small periplasmic turns in the periplasmic space (Fig 1) [14,38–40]. In the monomeric and dimeric forms, the C-terminal domain is located within the periplasm, where, for example, it is predicted to interact with Braun's lipoprotein and peptidoglycan [41–43]. In the dimeric form, the four extracellular loops are stabilized by the dimerization interface, consisting of interactions between extracellular loops 1, 2 and 4 and salt bridges in the C-terminal domain [44]. Loop 1 becomes buried in the dimerization interface. Loops 2 and 3, but less so loop 4, interact with other membrane components, such as lipopolysaccharides [44]. The third conformation has the C-terminal domains integrated within the membrane, forming another eight transmembrane segments, with the OmpA monomer then having 16 β-sheets [38] and functioning as a single-domain 'large pore' structure [14]. The 'large pore' conformation is expected to be a minority conformation [14], with characterization in unilameller proteoliposomes concluding that 2–3% of the OmpA molecules were in the 16-strand conformation [45].

In our previous study of environmental *E. coli* isolates, we observed allelic variation in OmpA [46]. This variation was observed at ten positions (Fig 1): each of the four extracellular loops, the transmembrane segments leading to and away from loop 3, the transmembrane segment leading away from loop 4, the at three locations within the C-terminal domain. Some variation in the N-terminal domain had been previously noted [27,47,48], though variation in the C-terminal domain had not. Specifically, we observed sequence variation at residues 175, 203 and 251. Within the 16-strand conformation [38], these each occur within a periplasmic

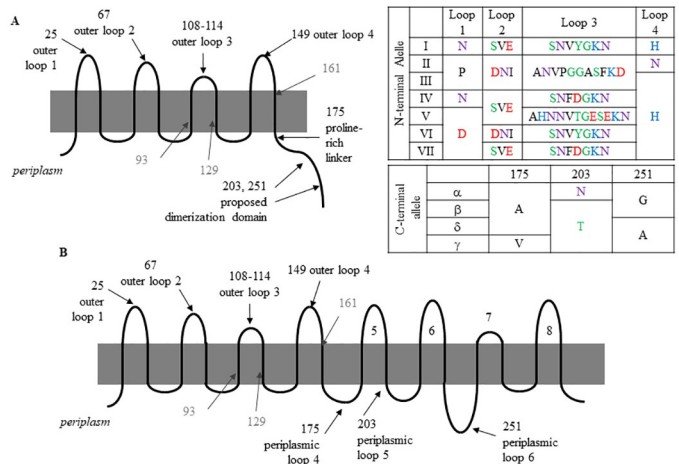

**Fig 1. Overview of OmpA structure and points of allelic variation as previously described** [46]. Residues 93, 129 and 161 (shown in gray) are not considered as part of the allele classification system used here. (A) The 8-strand conformation, based on [39]. (B) The 16-strand 'large pore' conformation, based on [38]. Inset, naming convention for OmpA alleles characterized here. Colors indicate amino acid chemistry, with polar residues shown in green, neutral residues in purple, basic residues in blue, acidic residues in red and hydrophobic residues in black.

loop (Fig 1). Residue 175 falls within the proposed 'sorting signal' region required for integration of OmpA into the outer membrane [49] and residue 203 is within the sequence associated with OmpA dimerization [40]. None of the these three residues are within the bulge structure associated with transition from one OmpA conformation to another [42].

We classified these patterns according to the N-terminal domain (I–VII) and the C-terminal domain (α, β, γ, δ), resulting in more than 20 distinct alleles [46]. Most K-12 *E. coli* strains encode allele (I, α), and this allele was the most abundant in our environmental isolate collection, accounting for 29.5% of strains. Expression of seven representative alleles (I, α), (II, α), (III, γ), (IV, β), (V, α), (VI, α), and (VII, β) in the same genetic background resulted in significantly different attachment to corn stover particles [46]. These variations in sequence were also noted by a 2019 characterization of adherent-invasive *E. coli* (AIEC), with the conclusion at sequence variation at position 175 –referred to as position 200 by Camprubi-Font et al–is associated with pathotype and with the adhesion index for intestinal epithelial cells [50]. A 2020 analysis of extraintestinal *E. coli* (ExPEC) characterized sequence variations in OmpA according to ExPEC type, and correlated these variations with phylotype, though a different classification system was used for the OmpA alleles [51]. This ExPEC analysis also included a consideration among ExPEC subtypes: NMEC (as defined above), uropathogenic *E. coli* UPEC), and avian pathogenic *E. coli* (APEC).

The impact of OmpA sequence variation, especially loops 2 and 3, on various aspects of *E. coli* physiology has been investigated. For example, identification of *ompA* mutants resistant to phage and colicin found that almost all mutations were within the loop 2 and 3 region [24]. Through application of computational analysis, Krishnan et al. (2014) reported that extracellular loops 1, 3 and 4 of OmpA are involved in NMEC's invasion of macrophage cells [52]. Hsieh et al. (2016) found that short peptides composed of one or two of OmpA's four extracellular loops protected mice from death after NMEC infection [53]. Singh et al's observation that the OmpA C-terminal domain generated a murine immune response indicates that at least some portions of the C-terminal domain are exposed on the microbial surface [54].

Here, we aimed to further characterize both the physiological implications and distribution of these OmpA alleles. Specifically, we investigate the extent to which different OmpA

sequences affect cell surface properties such as hydrophobicity, electrostatic charge, biofilm formation, and growth in the presence of inhibitors, when expressed in identical genetic backgrounds. We also compared the OmpA allele distribution among environmental isolates and ExPEC isolates, assessed the correlation with phylotype, and performed a phylogenetic analysis across representative members of the *Enterobacteriaceae* family.

# Materials and methods

## 2.1 Bacterial strains

Construction of the strains used here was previously described [46]. An *ompA* deficient MG1655 strain was used as the chassis for expressing seven distinct OmpA alleles. Each allele was cloned into the pGEN-MCS low copy plasmid, which can be well-maintained for generations without selective pressure [55,56], with expression driven by the MG1655 *ompA* promoter. The growth medium was supplemented with 100 μg ml$^{-1}$ ampicillin.

## 2.2 Zeta potential and hydrophobicity

Cells were grown to early stationary phase ($OD_{600}$ = 1.0–1.5) in M9 medium [57], initial pH 7.4, with 0.4% (wt/vol) glucose at 37˚C, 250 rpm. After harvesting by centrifugation at 4˚C, $4,000 \times g$ for 15 minutes, cells were washed and diluted with $CaCO_3$ solution at pH 8.0 and with ionic strength 10mM. Measurements were performed as described previously [58]. Briefly, zeta potential was measured at room temperature using a Zetasizer Nano-ZS with cells diluted in $CaCO_3$ solution to $OD_{600}$~0.1. Each technical replicate was assessed over 12 runs.

Hydrophobicity was measured using the microbial adhesion to hydrocarbon (MATH) method, in which cells are contacted with dodecane and partitioning of cells into the aqueous and organic phases quantified by $OD_{546}$.

Cell hydrophobicity (%) = 100* (initial $OD_{546}$ –aqueous phase $OD_{546}$)/aqueous phase $OD_{546}$ (1) The hydrophobicity measurement was modified slightly through the use of a multi-tube vortexer (Thermo Fisher Scientific Inc., Waltham, MA, USA) at 500 rpm for 10 minutes to homogenize the aqueous and organic phases instead of the previous method of vortexing at maximum speed for 2 minutes using a standard vortexer.

## 2.3 Biofilm assessment

The biofilm formation assay was conducted as previously reported [59]. Briefly, cells were grown in M63 minimal medium with 0.2% (wt/vol) glucose and 0.02% (wt/vol) casamino acids in non-cell-treated polystyrene 96-well microtiter plates (Falcon 351172) for 30 hours at 30˚C without agitation. These microplates were then agitated for 5 min at medium speed (linear shaking at 493 cpm, 4mm) using a microplate shaker (Biotek) to release the loosely attached cells and the $OD_{620}$ were measured. Wells were decanted and submerged in double distilled water three times to remove the free cells. After air drying at room temperature, wells were stained with 130 μl of 1% (wt/vol) crystal violet in ethanol for one hour at room temperature. Then 130 μl of 30% (vol/vol) acetic acid was added to each well. The plate was incubated at room temperature for 5 minutes, agitated at high speed for 5 minutes, and $OD_{570}$ was measured. The biofilm formation index was calculated as $OD_{570nm}/OD_{620nm}$.

## 2.4 Protein sequence analysis

Previously-reported OmpA sequences were used as exemplars for each of the seven alleles. Since our binning criteria are based on the amino acid sequence, we refer to OmpA alleles using protein-specific nomenclature. The charge of proteins and protein segments at pH 8.0

were estimated using Protein Calculator 3.4 (http://protcalc.sourceforge.net/). The hydrophobicity index was estimated using GPMAW lite (https://bio.tools/gpmaw_lite). Alignment and phylogenetic reconstructions were performed with ETE v3.1.1 [60], implemented on GenomeNet (https://www.genome.jp/tools/ete/). The tree was assembled using fasttree with slow NNI and MLACC = 3 [61]. Values at each node are the Shimodaira-Hasegawa (SH)-like local support.

## 2.5 Tolerance

Single colonies were inoculated into 2 ml LB media with 100 μg ml$^{-1}$ ampicillin and grown for 4 hours at 37 ˚C and 250 rpm. 60 μl of these log-phase cultures were used to inoculate 3 ml MOPS 2% w/v dextrose (100 μg ml$^{-1}$ ampicillin) at 37 ˚C and 250 rpm for ~19h. 150 μl of this seed culture was centrifuged at 6000 rpm for 5 min and the cell pellets were washed with 500 μl of PBS. Washed cells were centrifuged again as indicated and diluted into 1 ml of fresh media, containing various inhibitors where indicated. Inhibitors included 2.8% and 4.2% v/v ethanol, 1% v/v butanol, 1% v/v isobutanol, 0.2% v/v hexanol, 100 mM acetate at pH 7.00, as well as only MOPS 2% w/v dextrose media adjusted to pH 5.00 or pH 6.00.

Cells were grown in a 96-well plate with each well containing 200 μl of diluted culture, with three technical replicates for each of three biological replicates. The 96-well plate was placed in an Eon™ microplate spectrophotometer (BioTek), with Gen5 2.05 software, for 15 h at 205 cpm. The incubation temperature was 42˚C or 37˚C, as indicated. Optical density (550 nm) was measured every 10 minutes.

Growth curves were linearized by obtaining the natural logarithm of the $OD_{550}$ value divided by the initial $OD_{550}$ measurement for each sample. The duration of the log-phase was determined for by comparing the slopes between each time point of the linearized plot, where a decrease in slope was interpreted as the end of log-phase. Specific growth rates were calculated during the observed log-phase and standard deviation of the slope was estimated from the linearized plot for use as the confidence interval of the estimated growth rate.

## 2.6 Sensitivity to Neutrophil Elastase

Quantification of cell killing by neutrophil elastase followed a previously described procedure [19]. Human neutrophil elastase (Innovative Research, USA) was prepared in 10 mM PBS (pH = 7.4). Cells were grown in LB at 37˚C, 250 rpm to $OD_{600}$ ~ 0.5 and harvested by centrifugation at 4˚C, 4,000 × g for 15 minutes. Pelleted cells were washed twice with 10 mM PBS (pH = 7.4) and resuspend in PBS to $OD_{600}$ = 0.5. Then 2 μl of cell suspension and 1 μl LB were added to 97 μl of 2 μM neutrophil elastase solution and incubated at 37˚C. For the non-treated control, cells were suspended in PBS with no neutrophil elastase. Colony forming units (CFU) were quantified by serial dilution and plating after four hours of incubation.

## 2.7 Statistical analysis

Experiments typically used at least three biological replicates, with error bars indicating the standard deviation. Propagation of error calculations were performed when appropriate. The observed experimental precision was used to select an appropriate number of significant digits when reporting numerical values. LINEST (Excel) was used for correlations. Correlations were judged as significant if the 95% confidence interval of the slope did not span zero. Statistical analysis was performed using single-factor ANOVA, two-tailed student's t-test, and Fisher's exact test. Values are only described as differing if statistical criteria have been satisfied. Unless stated otherwise, a P-value cut-off of 0.005 was used as the criteria for significance.

## Results

In order to distinguish possible changes in cell properties caused by OmpA from those due to other genetic factors, we characterized seven exemplar OmpA alleles in a single genetic background. Specifically, we deleted the *ompA* gene from the MG1655 genome and individually cloned seven distinct *ompA* alleles into plasmid pGEN-MCS, with expression driven by the MG1655 *ompA* promoter [46]. pGEN-MCS is derived from pGEN222 [55,56] and uses the p15A origin of replication, which was determined to be maintained at 18 plasmid copies per chromosome equivalent during exponential growth [62]. These constructs were then used to explore the possible effect of OmpA allelic variation on cell properties.

### OmpA Allelic variation impacts cell surface properties

Cell surface charge has been linked with tolerance of disinfection agents, such as benzalkonium chloride [63] and is also an intriguing target for engineering, such as through the use of conjugated oligoelectrolytes [64]. Given the fact that many of the variations in these OmpA alleles involve charged amino acids, it is possible that these allelic variations may be sufficient to perturb the cell surface charge. Consistent with this expectation, expression of each of these seven OmpA alleles in the same genetic background led to variation in zeta potential (single factor ANOVA $p = 8 \times 10^{-13}$) (Fig 2A). Specifically, expression of the (IV, β) allele was associated with a less negative zeta potential value than all other alleles. The (III, γ) allele also resulted in a statistically unique value, intermediate to (IV, β) and the other five alleles. The observed range of approximately -5 to -33 mV is consistent with the range of -7 to -40 mV reported in our previous characterization of a set of 78 environmental *E. coli* isolates [65] and the range of -4.9 to -33.9 mV reported for 22 *E. coli* isolates [66]. Our observed charge of -29 mM for MG1655 Δ*ompA* expressing the (I, α) allele (Fig 2A) is consistent with the previously-reported charge of -24 mV for K-12 strain JM109, which also encodes the (I, α) allele [66].

Cell surface hydrophobicity is an intriguing engineering target for improving bioproduction. Increased hydrophobicity has been associated with increased production of short-chain fatty acids, but there are few known genetic targets for tuning hydrophobicity [3,8,9,67]. As with the observed variation in zeta potential, the various OmpA alleles were associated with variations in cell surface hydrophobicity (single factor ANOVA $p = 1 \times 10^{-12}$) (Fig 2B). Expression of the (VI, α) allele resulted in a value that was statistically lower than all other alleles. Patterns (I, α), (II, α) and (V, α) formed a statistically distinct cluster with a lower hydrophobicity than alleles (VII, β), (III, γ) or (IV, β).

While the zeta potentials observed for MG1655 Δ*ompA* expressing different OmpA alleles was consistent with the range of values observed for 78 environmental isolates, the range of observed hydrophobicity values for these MG1655 derivatives was much smaller than the range observed for the environmental isolates. Specifically, the cell surface hydrophobicity ranged from 1–90% for the environmental isolates [65], but ranged only from 1–12% for the engineered MG1655 strains (Fig 2B). This substantially dampened range of hydrophobicity values emphasizes the contribution of other cell features to this cell surface property.

Microbial biofilms are due to complex genetic factors [68–70]. OmpA, like many other proteins, has been implicated in *E. coli* biofilm formation [71,72]. Here, we investigated the influence of OmpA allelic variation on biofilm formation. Strains expressing the various OmpA alleles fell into two distinct groupings of biofilm formation (single-factor ANOVA $p = 2 \times 10^{-13}$) (Fig 2C). Strains expressing alleles (IV, β), (VII, β) and (III, γ) formed, on average, 10-fold less biofilm than all of the α alleles. This clear demarcation between alleles with the α C-terminal domain allele relative to those with the β and γ alleles suggests that the C-terminal domain

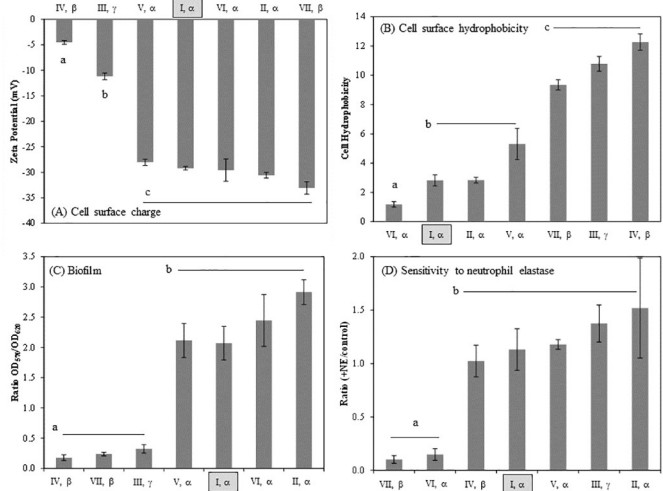

**Fig 2. *E. coli* cell surface properties vary (one-way ANOVA, p < 0.005) according to OmpA allele.** OmpA variants were expressed in MG1655 *ΔompA* from the pGEN-MCS plasmid using the MG1655 *ompA* promoter, as previously described [46]. Statistically distinct groups are designated with different lowercase letters (two-tailed students t-test, p < 0.005). Error bars indicate the standard deviation. K-12 *E. coli* expresses the (I, α) allele and this allele is indicated in each figure. (A) Cell surface charge, measured in $CaCO_3$ solution pH 8.0 with 10 mM ionic strength. (B) Cell surface hydrophobicity, assessed by partitioning cells between aqueous and organic phases. (C) Propensity for biofilm formation in defined media containing glucose and casamino acids at 30°C. (D) Sensitivity to killing by neutrophil elastase (NE), assessed by quantification of CFUs four hours after treatment relative to a sham control.

strongly influences this cell behavior. Or, this difference may be due to incompatibility between the K12 host strain and non-K12 C-terminal domain variants.

Neutrophil elastase is one of several mammalian proteins known to interact with OmpA [19,52,73]. Previous characterization of *E. coli* strain RS218 demonstrated that OmpA is an essential target for cell-mediated killing by neutrophil elastase [19]. Strain RS218 is associated with neonatal meningitis and encodes the (I, δ) OmpA allele [74], which was is not among our seven representative alleles. We observed neutrophil elastase-mediated killing in MG1655 *ΔompA* upon expression of only two of the seven exemplar alleles: (VII, β) and (VI, α). This killing was evidenced by a decrease (p < 0.005) in CFUs relative to the corresponding untreated control (Fig 2D). For the other five alleles, there was no detectable killing effect of the neutrophil elastase. Perhaps it is notable that alleles (VII, β) and (VI, α) are the only two alleles with both the D variation in loop 1 and a 7-residue sequence in loop 3, as opposed to the longer sequence observed in alleles II, III and V.

## Correlation and groupings among cell surface properties

Having observed variation in these cell surface properties in response to allelic variation in OmpA, we looked for associations between these metrics, without consideration of the underlying amino acid sequences. Associations between metrics were evaluated based on the 95% confidence interval of the slope of a linear best fit. Specifically, a 95% confidence interval that did not span zero was used as evidence of a significant association.

Across the seven exemplar alleles characterized here, there was a positive correlation between hydrophobicity and zeta potential and a negative correlation between hydrophobicity and biofilm formation (Fig 3A and 3B). A group of three alleles was observed to have statistically similar zeta potential and hydrophobicity: (I, α), (II, α) and (V, α). Two other groups showed statistically similar hydrophobicity and biofilm values: (I, α), (II, α) and (V, α); and

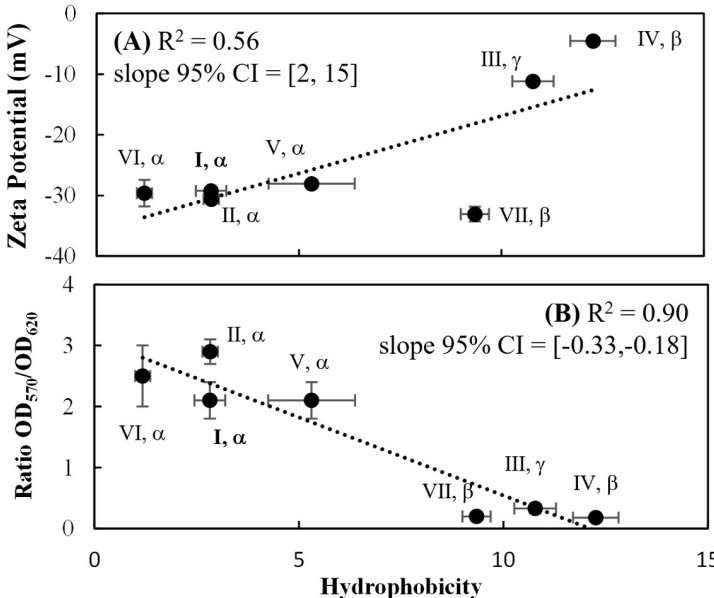

**Fig 3. Correlation between cell surface properties, judged by the 95% confidence interval (CI) of the estimated slope.** (A) Association between cell surface charge (zeta potential) and hydrophobicity. (B) Association between propensity for biofilm formation and hydrophobicity. The (I, α) allele, encoded by K12 *E. coli* strains, is bolded for reference.

(VII, β), (III, γ) and (IV, β). Other combinations of metrics did not meet the criteria for statistical significance. It should be noted that (I, α), (II, α), and (V, α) have statistically similar values for all four of these cell surface properties. There are also two pairs of alleles that statistically differ for each of the four metrics: (III, γ) and (VI, α); and (IV, β) and (VI, α).

## Alignment between sequence variations and cell surface properties

It is expected that the observed variation in surface properties would correspond to the known sequence variations in OmpA. For example, the propensity for biofilm formation parses cleanly with the C-terminal domain allele identity (Fig 2C). To be more specific, alleles presenting an N at position 203 have a high propensity for biofilm formation, while those presenting a T have a low propensity for biofilm formation. Other properties are not as readily explained by sequence variation. For example, alleles (I, α) and (II, α) are statistically indistinguishable across all four cell surface properties, despite the fact that these two alleles differ at each of the four N-terminal domain outer loops. In contrast, (IV, β) and (VII, β) differ only by a single residue within loop 1, and yet have statistically differing cell surface charge and sensitivity to neutrophil elastase (Fig 2D).

The seven exemplar alleles were binned according the sequence at distinct positions within the OmpA protein. For example, alleles (I, α), (IV, β), (V, α) and (VII, β) each encode an SVE at loop 2, while the other three alleles encode DNI (Fig 1). The analysis identified only one sequence variation as correlated with any of the four cell surface properties. Specifically, the variation at position 203 correlated with hydrophobicity (p = 0.0015) and, as noted above, biofilm formation (p = 0.0003). There were no detectable trends for any of the N-terminal domain sequence variations.

Quantitative modeling of the expected charge and hydrophobicity index for the various loops also showed poor alignment with the experimental observations (Table 1). Specifically,

**Table 1. Sequence-based estimation of (A) peptide charges and (B) hydrophobicity index. K12 E. coli strains express the (I, α) allele, which is bolded here for reference.**

(A)

| *Charge* | | IV, β | III, γ | V, α | **I, α** | VI, α | II, α | VII, β |
|---|---|---|---|---|---|---|---|---|
| experimental group (Fig 2A) | | a | b | | | c | | |
| | Loop 1 | -2.4 | -2.4 | -3.4 | -2.4 | -3.4 | -2.4 | -3.4 |
| | Loop 2 | -0.5 | | | | | | |
| | Loop 3 | -0.5 | +0.5 | -1.5 | +0.5 | +0.5 | +0.5 | -0.5 |
| | Loop 4 | -1.5 | | | | | | |
| Σ N-terminal domain loops | | -4.9 | -3.9 | -6.9 | -3.9 | -4.9 | -3.9 | -5.9 |
| | Loop 5 | -0.5 | | | | | | |
| | Loop 6 | -1.5 | | | | | | |
| | Loop 7 | -1.5 | | | | | | |
| | Loop 8 | -1.0 | | | | | | |
| Σ C-terminal domain loops | | -4.5 | | | | | | |
| Σ N- and C-terminal domains extracellular loops | | -9.4 | -8.4 | -11.4 | -8.4 | -9.4 | -8.4 | -10.4 |

(B)

| *Hydrophobicity* | | VI, α | **I, α** | II, α | V, α | VII, β | III, γ | IV, β |
|---|---|---|---|---|---|---|---|---|
| experimental group (Fig 2B) | | a | b | | | c | | |
| | Loop 1 | -1.7 | -1.7 | -1.6 | -1.7 | -1.7 | -1.6 | -1.7 |
| | Loop 2 | -0.9 | -0.7 | -0.9 | -0.7 | -0.7 | -0.9 | -0.7 |
| | Loop 3 | -1.0 | -1.0 | 0.0 | -1.6 | -1.6 | 0.0 | -1.6 |
| | Loop 4 | -1.1 | -1.1 | -1.2 | -1.1 | -1.1 | -1.1 | -1.1 |
| Σ N-terminal domain loops | | -4.7 | -4.5 | -3.6 | -5.1 | -5.1 | -3.6 | -5.1 |
| | Loop 5 | -0.9 | | | | | | |
| | Loop 6 | -0.9 | | | | | | |
| | Loop 7 | -1.1 | | | | | | |
| | Loop 8 | -0.5 | | | | | | |
| Σ C-terminal domain loops | | -3.4 | | | | | | |
| Σ N- and C-terminal domains extracellular loops | | -8.1 | -7.8 | -7.0 | -8.4 | -8.4 | -7.0 | -8.4 |

comparison of the experimental cell surface charge values (Fig 2A) with the sum of the estimated N-terminal domain loop charges did not show a significant correlation (p = 0.6). Correlation of experimental hydrophobicity measurements and the sum of the estimated N-terminal domain loop hydrophobicity values was similarly poor (p = 0.7). However, these estimated values do not account for changes in loop exposure in the dimeric relative to the monomeric 8-strand conformation, or for changes that would occur upon shifting from the 8-strand to the 16-strand conformation. It is possible that the sequence variations impact the distribution of OmpA among the three different structural conformations and that this would need to be accounted for when making sequence-based predictions of the charge and hydrophobicity of exposed residues.

## OmpA Alleles impact tolerance

Thus far, our analysis has been restricted to features of the microbial cell surface. OmpA plays a role in transporting material into and out of the cell and has previously been implicated in tolerance to various chemical stressors, such as phenylpropanoids and phenol [75,76]. Here, each of the OmpA alleles were characterized in terms of their ability to confer tolerance to various stressors relevant to bio-production and food spoilage: ethanol, butanol, isobutanol, hexanol, acetate, pH 5.00, pH 6.00 and 42˚C.

MG1655 $\Delta ompA$ expressing each of the seven distinct exemplar OmpA alleles was grown in the presence of the various stressors, with specific growth rate in defined minimal glucose medium was used as the indicator of tolerance (Fig 4). Specific growth rate was used here as a metric of tolerance due to its relevance to bioproduction. Variation in growth rate in response to OmpA amino acid sequence was observed only for 42˚C and for acetate stress (single-factor ANOVA p values = $9x10^{-5}$ and $5x10^{-4}$, respectively).

At 42˚C, the (IV, β) allele is prominent in terms of its high specific growth rate (Fig 4A). With the statistical criteria used here (two-tailed t-test, p < 0.005), (IV, β) differed only from (VII, β). But it should be noted that comparison of the (IV, β) growth rate to each of the other alleles had p-values < 0.05. In the presence of acetate, strains expressing (IV, β) and (VI, α) had a higher specific growth rate than the strain expressing the (V, α) allele.

As with the cell surface properties, the specific growth rates of cells expressing the (I, α), (II, α) and (V, α) alleles are statistically indistinguishable. While (IV, β) and (VI, α) and (IV, β) and (III, γ) always differed in their cell surface property values, no differences in their specific growth rates were observed here, though cell surface properties were measured only in the baseline condition, not in the presence of these stressors.

## Allele distribution among isolate types

We previously described the collection and characterization of 78 environmental *E. coli* isolates and determined the OmpA allele distribution [46,58,65]. Camprubi Font et al described sequence variation for OmpA, OmpC and OmpF among 13 adherent-invasive *E. coli* (AIEC) isolates and 30 non-AIEC mucosa-associated *E. coli*, but did not explicitly define alleles [50]. Nielsen et al subjected 399 ExPEC isolates to *ompA* sequencing [51], consisting of APEC (n = 171), UPEC (n = 148) and NMEC (n = 80) isolates [51]. This ExPEC characterization used a different binning process for OmpA alleles than the process described in our prior work [46]. Both binning processes consider sequence variations at outer loops 1–4, at position 175 (proline-rich linker) and at positions 203 and 251 within the C-terminal domain. Nielsen et al's binning also accounts for positions 93, 129 and 161 (Fig 1), each of which occur in transmembrane segments and were also described in [46].

We binned the OmpA sequences from Camprubi Font's non-AIEC and AIEC isolates and Nielsen's APEC, UPEC and NMEC isolates using the criteria presented in our prior work and displayed in Fig 1, consisting of a total of 412 ExPEC OmpA sequences. Here we report that the distribution of OmpA alleles among these two previously-described groups–environmental isolates [46] and ExPEC isolates [50,51]–have different OmpA allele distributions (Fig 5, Table 2). Specifically, the (I, α) allele is enriched in environmental isolates relative to ExPEC, while the (I, δ) allele is enriched in ExPEC isolates relative to the environmental isolates (Fisher's exact test, $p<1x10^{-5}$). This comparison of OmpA allele distribution among environmental isolates and ExPEC isolates has not been previously described.

Nielsen et al used the Chi square test of homogeneity with p < 0.05 as criteria for significance and identified alleles that were enriched among ExPEC types [51]. The sample sizes described here are sufficiently large for the Chi square test, using criteria described by [77]. Nielsen's analysis concluded: A1 (I, δ), C4 (III, γ), and D1 (IV, δ) were enriched in UPEC; B2 (III, α), D30 (IV, β), and F2 (II, α) were enriched in APEC; and C1 (III, δ) was enriched in NMEC. Camprubi-Font et al determined that the amino acid coded at the position described here as 175 was significantly different between AIEC and non-AIEC intestinal mucosa isolates [50]. This variation is binned here as the γ C-terminal allele (Fig 1).

Here we used our prior binning criteria and Fisher Exact Test with a cutoff of $p < 1x10^{-5}$ (Table 2, S1A Fig). We observed enrichment of (I, δ) in NMEC isolates relative to UPEC and

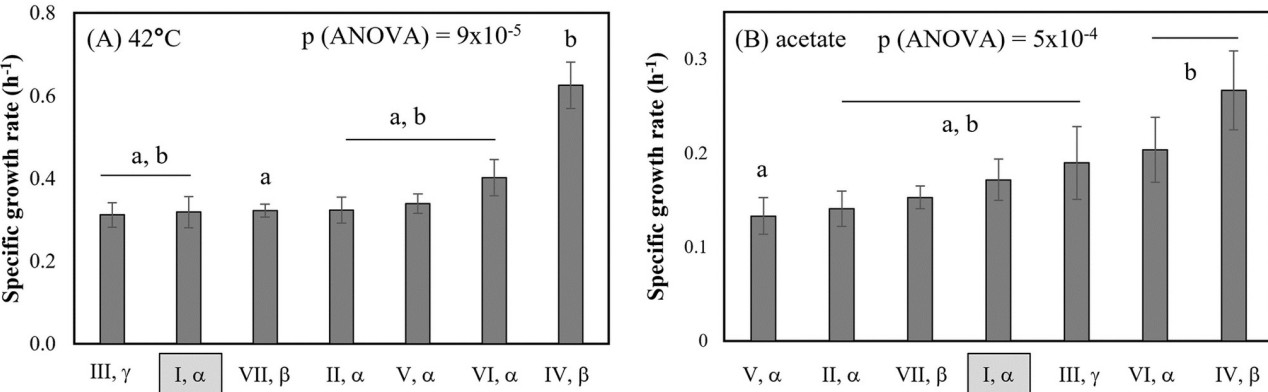

**Fig 4. Expression of various OmpA alleles is associated with changes in log-phase specific growth rate during growth in glucose minimal media at (A) 42˚C; and (B) 37˚C with 100 mM acetate (pH 7.0).** Other inhibitors did not meet the criteria of statistical significance. The (I, α) allele is indicated in each figure–this is the allele encoded in K-12 *E. coli* such as MG1655. Lowercase letters indicate statistically distinct values (two-tailed t-test, p<0.005).

APEC; enrichment of (III, α) in APEC relative to NMEC and UPEC; and enrichment of (III, γ) in AIEC and UPEC relative to APEC. These results are consistent with Nielsen's prior analysis and are more stringent and specify the statistical relationship between each of the three ExPEC subtypes. Comparison of the AIEC sequence distribution with APEC, NMEC and UPEC has not been previously described.

Finally, we evaluated the OmpA allelic distribution according to phylotype. This analysis includes 78 environmental isolates [46], 30 non-AIEC mucosal isolates [50], and 412 ExPEC isolates [50,51]. The dataset was trimmed to include only the 400 isolates with A (n = 45), B1

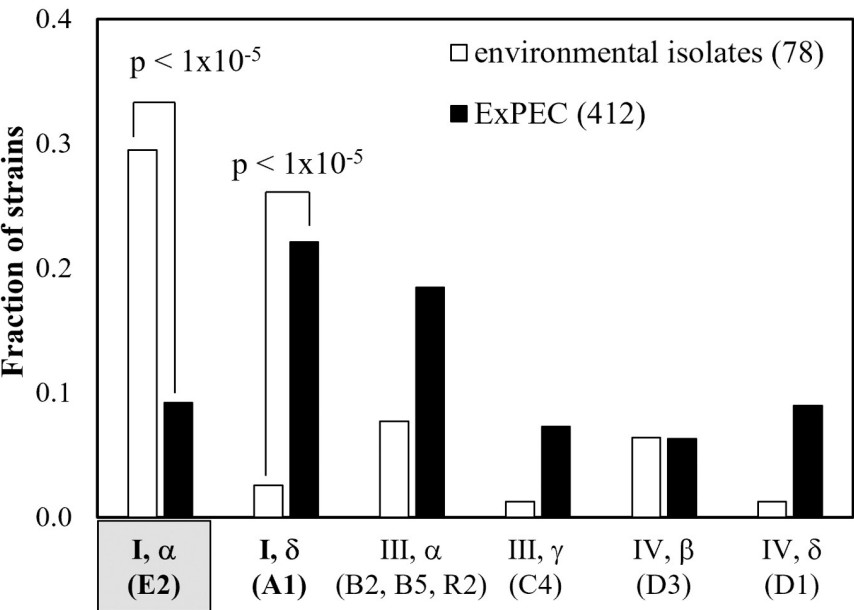

**Fig 5. The OmpA (I, δ) allele is enriched in ExPEC relative to environmental isolates.** Terms in parentheses below the x-axis labels are the naming system used by [51]. The previously-described distribution of OmpA among 78 environmental isolates [46] is compared to 412 ExPEC isolates [50,51]. The (I, α) allele is indicated with shading, as this is the allele encoded in K-12 *E. coli* such as MG1655.

**Table 2. Summary of enrichments of OmpA sequences among phylotypes.** This table summarizes the environmental vs ExPEC results shown in Fig 4 and describes comparisons among ExPEC types. The (I, α) allele is shaded, as this is the allele encoded in K-12 *E. coli* such as MG1655. No significant differences were observed among the 40 OmpA sequences in the 'other' category.

| Allele | Enrichments (p < 1x10$^{-5}$) |
|---|---|
| I, α (n = 57) | Environmental (29%) relative to ExPEC (9%)<br>Phylotype A (49%) and B1 (49%) relative to B2 (2%), F (0%)[1] |
| I, δ (n = 96) | ExPEC (22%) relative to environmental (3%)[2]<br>NMEC (53%) and UPEC (26%) relative to APEC (6%)[3]<br>Phylotype B2 (100%) relative to A (0%), B1 (0%) and F (0%)[1] |
| II, α (n = 53) | Phylotype B1 (91%) relative to A (6%), B2 (4%) and F (0%)[1] |
| III, α (n = 14) | APEC (39%) relative to NMEC (4%) and UPEC (5%)[3]<br>Phylotype A (57%) relative to B2 (7%)[1] |
| III, δ (n = 41) | *Fisher's Exact test across phylotypes has p < 1x10$^{-5}$, but no pairwise comparisons meet p < 1x10$^{-5}$ criteria*[1] |
| III, γ (n = 36) | AIEC (46%) and UPEC (16%) relative to APEC (0%)[3]<br>B2 (100%) relative to B1 (0%) |
| IV, β (n = 23) | Phylotype F (96%) relative to A (0%), B1 (0%) and B2 (4%)[1] |
| IV, δ (n = 40) | Phylotype B2 (100%) relative to B1 (0%)[1] |

[1]Phylotype distribution among APEC, UPEC and NMEC was presented by Nielsen et al [51] and AIEC by Camprubi Font [50]. Here we have applied Liao's binning criteria, added data for environmental isolates and performed statistical analysis (S1B Fig).

[2]The distribution of OmpA alleles among environmental isolates was presented by [46], AIEC sequences by [50]. APEC, UPEC and NMEC sequences were described and binned using different binning criteria than those used here [51]. Here we have applied Liao's binning criteria and compared the distribution across groups (Fig 5).

[3]Nielsen et al assessed the OmpA distribution among APEC, NMEC and UPEC subtypes, using Chi square analysis and p < 0.05. Here we added Camprubi Font's AIEC sequences, applied Liao's binning criteria, and used Fisher Exact test (p < 1x10$^{-5}$) (S1A Fig).

(n = 83), B2 (n = 229) or F (n = 43) phylotypes. This trimmed set consists of 47 environmental isolates (12%) and 317 of Nielsen's ExPEC isolates (79%) and 36 of Camprubi Font's mucosal isolates (9%). These were binned according to the OmpA allele, and alleles represented by less than 10 isolates were lumped together as "other" (n = 40). The remaining eight alleles in the dataset include: (I, α), n = 57; (I, δ), n = 96; (II, α), n = 53; (III, α), n = 36; (III, γ), n = 36; (III, δ), n = 41; (IV, β), n = 23; (IV, δ), n = 40. Fisher's Exact Test was applied with a criterion of p < 1x10$^{-5}$ and differences in distribution were observed for (I, α), (I, δ), (II, α), (III, α), (III, δ), (III, γ) (IV, β) and (IV, δ), as summarized in Table 2 (S1B Fig).

The new results presented here show that OmpA allele distribution differs for environmental isolates relative to ExPEC isolates (Fig 5). We have also shown that a pool of *E. coli* strains including both environmental and ExPEC isolates have differences in OmpA allele distribution according to phylotype (S1B Fig, Table 2), consistent with prior analyses of the ExPEC isolates [51]. Finally, it should be noted that the relative abundance of the (I, δ) allele significantly varied for each of the comparative metrics (Table 2): environmental vs ExPEC (Fig 5); among ExPEC types (S1 Fig); and among phylotype (S1B Fig).

## OmpA alleles in Gram-negative bacteria

OmpA is highly conserved among some bacteria in *Enterobacteriaceae* family [35,78]. Here, we present a comprehensive phylogenetic tree generated from the OmpA amino acid

sequences of representative members in the *Enterobacteriaceae* family. Twelve intestinal pathogenic *E. coli* isolates related to waterborne diseases were included to represent Enteroaggregative *E. coli* (EAEC), Enterotoxigenic *E. coli* (ETEC), Enteropathogenic *E. coli* (EPEC), Shiga toxin producing *E. coli* (STEC) and Enteroinvasive *E. coli* (EIEC). Nine strains of *S. flexneri*, *S. boydii*, *S. dysenteriae* and *S. sonnei* representing human pathogenic bacteria genus *Shigella*; three strains from genus *Escherichia*: *E. albertii*, *E. fergusonii* and *E. vulneris*; one of each following bacteria: *Salmonella enterica*, *Yersinia pestis*, *Klebsiella pneumoniae* and *Raoultella planticola* were also included in this tree. All sequences, except for our environmental isolates, were obtained from NCBI (Fig 6). To the best of our knowledge, this systematic comparison of OmpA sequence and allele identity across species has not been previously described.

The phylogenetic tree consists of three sub-clusters. The first and biggest cluster includes genera *Escherichia*, *Shigella* and *Salmonella*; the second contains *Klebsiella* and *Raoultella*; the third is *Yersinia*. Within the genus *Escherichia*, *E. vulneris* OmpA shares the least similarity with *E. coli*, even less than *Salmonella* and *Shigella* species. OmpA of *Shigella* and *E. coli* show a high degree of similarity. For example, the two strains of *Shigella flexneri* encode (II, α) and (I, α) alleles. This cluster is dominated by the α C-terminal domain variant, with the β, γ and δ variants being infrequently observed.

OmpA of *Klebsiella*, *Raoultella*, *Yersinia pestis* and *Salmonella enterica* are sufficiently distinct from the *E. coli* sequences such that they cannot be described in the classification system used here. There are reports of allelic variation of OmpA for both *Y. enterocolitica* [79] and *Y. ruckeri* [80]. Given the difference of (I, δ) relative abundance among *E. coli* isolates (Table 2), it is notable that this allele was not encoded by any of these representative members of the *Enterobacteriaceae* family.

Approximately 50 isolates with 100% identity match to residues 211–325 of our example (I, δ) variant were identified using SmartBLAST (S1 Table), nine of these isolates have N-terminal domain pattern (I). These (I, δ) alleles include two isolates from adherent invasive *E. coli* [81,82], an APEC isolate [83] and UPEC isolate UTI89 [84]. Comparison of the UTI89 genome with other *E. coli* genomes identified OmpC and OmpF, but not OmpA, as being subject to positive selection [84].

## Discussion

OmpA is an outer membrane protein with many previously-described biotechnological applications. It can be used as a carrier protein for surface display of foreign peptides for acting as biocatalyst, bio-chelate [85], and vaccine [86], as well as used for peptide library screening [87]. In this study, we found that seven distinct alleles of OmpA are associated with differences in zeta potential (Fig 2A), hydrophobicity (Fig 2B), biofilm formation (Fig 2C), sensitivity to neutrophil elastase (Fig 2D), growth in the presence of mild thermal stress (42˚C) (Fig 3A), and growth in the presence of acetate at neutral pH (Fig 3B). These results suggest that it may be possible to use OmpA as a means of further tuning of some of these properties. The (IV, β) allele is intriguing in that it is frequently distinguished from the other alleles: it is associated with the least negative cell surface charge, the highest hydrophobicity, and the highest specific growth rate at 42˚C and in the presence of acetate. In our characterization of *E. coli* attachment to corn stover, the (IV, β) variant was associated with the lowest attachment behavior [46]. The temperature-dependent structural transition of the C-terminal domain [42] may be related to the observed increase in specific growth rate at 42˚C for the (IV, β) allele.

In our analysis of environmental *E. coli* isolates, a correlation between cell surface charge and hydrophobicity was not observed [65]. Similarly, Li and McLandsborough's characterization of various *E. coli* isolates did not detect a significant relationship between cell surface

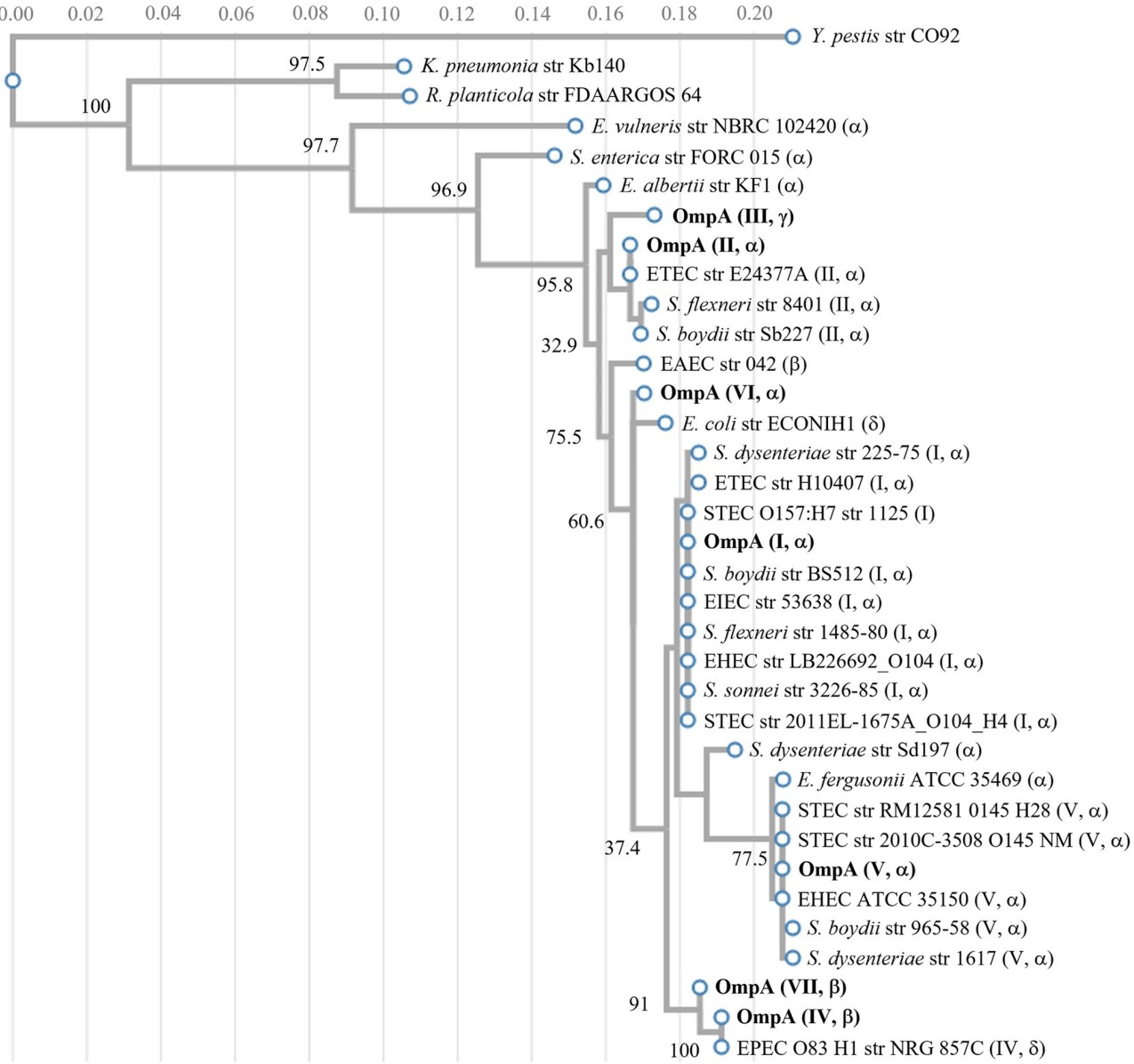

**Fig 6. Phylogenetic distribution of the amino acid sequence of the seven exemplar OmpA alleles characterized here amongst representative members of the *Enterobacteriaceae* family.**

charge and hydrophobicity [66]. Here, when the seven alleles were all expressed in the same genetic background, a significant relationship was observed (Fig 2D).

The poor alignment between the sequence of our OmpA alleles and the observed trends in cell surface charge and hydrophobicity are concerning (Table 1). However, previous characterization of OmpA alleles concluded that the interaction between OmpA and lipopolysaccharide has a degree of strain specificity [48]. For example, a K-12 *E. coli* strain expressing its native OmpA showed more than double the interaction with erythrocytes with K-12 LPS than the same strain expressing OmpA from each of two clinical isolates. OmpA is known to engage in site-specific interactions with other membrane components, such as RcsF [88–90] and Braun's

lipoprotein [41]. Perturbations of these interactions due to changes in OmpA sequence could have secondary effects that influence cell surface properties. The lack of correlation between our analysis of the OmpA sequence and the observed experimental trends could also be due to differing distribution between the three OmpA conformations. Difference in the abundance of different OmpA alleles is also a possible confounding factor not controlled for here. It has been previously reported that OmpA protein abundance did not differ between AIEC and non-AIEC grown in Mueller Hinton broth, even though these two groups significantly differed in OmpA sequence [50].

Power et al noted *ompA*'s allelic variation and defined two alleles: *ompA1* and *ompA2* [47]. With the classification system used here, *ompA1* encompasses outer loop alleles I, IV and VII and *ompA2* encompasses outer loop alleles II and III. Power et al performed PCR-based screening of 524 *E. coli* isolates, including human (clinical and fecal) and non-domesticated Australian mammals [47]. Similar to the results presented here and by [51], significant differences were observed in allele distribution according to phylotype and source. Power's analysis also concluded that isolates with the *ompA2* allele had decreased sensitivity to lysis by bacteriophage [47]. Li and McLandsborough observed a drastic difference in cell surface charge between the K-12 strain JM109 and the O157:H7 serotype strain ATCC 43895, with values of approximately -24 mV and -6 mV, respectively [66], even though both strains encode the (I, α) allele.

Our analysis of the distribution of OmpA alleles determined that the (I, δ) allele is enriched in ExPEC isolates relative to environmental isolates (Fig 5). Consistent with the observations of [51], variations in OmpA allele frequency among APEC, NMEC and UPEC isolates were observed (S1 Fig). The phylotype analysis previously performed for ExPEC isolates [51] was expanded to include environmental isolates, with significant enrichments observed (Table 2). Most notably, the (I, δ) allele, in addition to being enriched in ExPEC isolates relative to environmental, was also enriched in NMEC relative to APEC and UPEC, and in the B1 phylotype relative to A, B2 and F.

A recent characterization of commensal *E. coli* strains from colorectal cancer concluded that within the B2 phylogenetic group, OmpA abundance was significantly increased in the cancer-associated isolates relative to the control group [91]. The OmpA sequences from these isolates are not available, but the enrichment of the OmpA (I, δ) and (IV, δ) alleles within the B2 phylotype (Table 2, [51]) is possibly relevant to this finding. It is tempting to view the repeated appearance of the C-terminal domain δ allele in these analyses as further evidence that the OmpA topological model is incomplete.

Finally, we explored OmpA variation across the *Enterobacteriaceae* family, noting extensive sequence similarity between *E. coli* and *S. dysenteriae* (Fig 6). Shiga toxin encoding genes are reported to be transmitted from *S. dysenteriae* to *E. coli* through bacteriophage [92]. The resulting STEC, such as O157:H7 and O104:H4, cause severe food-borne disease. Our finding about *E. coli* and *S. dysenteriae* having similar OmpA sequences suggests that OmpA may be horizontally transmitted. Sequence diversity of OmpA, particularly in loop 3, of *Yersinia ruckeri* was concluded to be independent of phylogeny [80]. This independence was interpreted as evidence of horizontal gene transfer and selective pressure on the *ompA* gene. We observed here that OmpA variants, particularly the (IV, β) allele, can provide a growth advantage in conditions related to food spoilage and bioproduction (Fig 3). Allelic variation in outer membrane proteins have been noted as means of clone expansion, such as with the major outer membrane protein PorA in *Campylobacter jejuni* [93]. The high-level variation of OmpA extracellular loops can possibly be attributed to selective pressure by the mammalian immune system. Using the flagellin gene encoding the H antigen as a model allelic system, Wang et al. (2003)

concluded that a selective advantage of only 0.1% is sufficient for maintaining a specific niche [94].

## Supporting information

**S1 Fig. Distribution of OmpA alleles in *E. coli* isolates.** Fisher's Exact test was used to identify significant differences (two tailed, $p < 1 \times 10^{-5}$), as summarized in Table 2 in the main text. Bold font indicates alleles with significant differences, described in Table 2. Terms in parentheses below the x-axis labels are the naming system used by Nielsen et al. (A) Distribution of OmpA alleles among ExPEC isolates. Bold font indicates significant differences. (B) Phylotype distribution among the environmental and ExPEC isolates. Data is only shown for A, B1, B2 and F phylotypes. Color coding indicates outer loop allele (I, blue; II, green; III, gold; IV, red) and fill indicates C-terminal domain (α, solid; β, dotted; γ, horizontal stripes; δ, diagonal stripes). The (I, α) allele is indicated in each figure–this is the allele encoded in K-12 *E. coli* such as MG1655.
(PDF)

**S1 Table. Isolates with 100% sequence identity with residues 211–325 of the (I, δ) allele, as identified via Smart BLAST.**
(PDF)

**S1 File. *E. coli* OmpA amino acid sequences.**
(DOCX)

**S2 File. Raw experimental data for all figures and tables.**
(XLSX)

## Acknowledgments

Iowa State University is located on the ancestral lands and territory of Ioway Nation. We wish to recognize our obligations to this land and to the people who took care of it, as well as to the 17,000 Native people who live in Iowa today.

## Author Contributions

**Conceptualization:** Chunyu Liao, Miguel C. Santoscoy, Chiron Anderson, Michelle L. Soupir, Laura R. Jarboe.

**Data curation:** Chunyu Liao.

**Formal analysis:** Chunyu Liao, Miguel C. Santoscoy, Julia Craft, Chiron Anderson, Laura R. Jarboe.

**Funding acquisition:** Michelle L. Soupir, Laura R. Jarboe.

**Investigation:** Chunyu Liao, Miguel C. Santoscoy, Chiron Anderson, Laura R. Jarboe.

**Methodology:** Chunyu Liao, Miguel C. Santoscoy, Julia Craft, Chiron Anderson.

**Project administration:** Michelle L. Soupir, Laura R. Jarboe.

**Supervision:** Michelle L. Soupir, Laura R. Jarboe.

**Validation:** Chiron Anderson.

**Visualization:** Chiron Anderson.

**Writing – original draft:** Chunyu Liao.

**Writing – review & editing:** Chunyu Liao, Miguel C. Santoscoy, Michelle L. Soupir, Laura R. Jarboe.

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
