## [Decision Letter · Decision Letter 0]

1 Aug 2022

PONE-D-22-18140Allelic Variation of Escherichia coli Outer Membrane Protein A: Impact on Cell Surface Properties, Stress Tolerance and Allele DistributionPLOS ONE

Dear Dr. Jarboe,

Thank you for submitting your manuscript to PLOS ONE. After careful consideration, we feel that it has merit but does not fully meet PLOS ONE’s publication criteria as it currently stands. Therefore, we invite you to submit a revised version of the manuscript that addresses the points raised during the review process.

Both reviewers were very positive about your paper but they have made some suggestions, which I think are valid and would improve the paper.

We look forward to receiving your revised manuscript.

Kind regards,

Olivia Steele-Mortimer, Ph.D.

Academic Editor

PLOS ONE

Journal Requirements:

"Funding for this work was provided by the National Science Foundation grants CBET-1236510 and CBET-1604576, and the Unites States Department of Agriculture National Institute of Food and Agriculture, award number 2017-6702-26137. The funders had no role in study design, data collection and interpretation, or the decision to submit the work for publication."

"Funding for this work was provided by the National Science Foundation (https://www.nsf.gov/) grants CBET-1236510 (MLS) and CBET-1604576 (LRJ), and the Unites States Department of Agriculture National Institute of Food and Agriculture (https://www.nifa.usda.gov/), award number 2017-6702-26137 (LRJ). The funders had no role in study design, data collection and analysis, decision to publish, or preparation of the manuscript."

Reviewers' comments:

Reviewer's Responses to Questions

**Comments to the Author**

1. Is the manuscript technically sound, and do the data support the conclusions?

Reviewer #1: Yes

Reviewer #2: Yes

2. Has the statistical analysis been performed appropriately and rigorously? 

Reviewer #1: Yes

Reviewer #2: Yes

3. Have the authors made all data underlying the findings in their manuscript fully available?

Reviewer #1: Yes

Reviewer #2: Yes

4. Is the manuscript presented in an intelligible fashion and written in standard English?

Reviewer #1: Yes

Reviewer #2: Yes

5. Review Comments to the Author

Reviewer #1: In this manuscript the authors describe the OmpA variants found in ExPEC strains and their properties (membrane charge, hydrophobicity, formation of biofilm, etc.). Here are some suggestions and comments that may improve the manuscript.

1. The introduction should also give a small description of ExPEC, and what other groups form them (APEC, UPEC, NMEC). Those are mentioned during the manuscript, but a short description or definition at the introduction would help the reader. Also, why were those 7 alleles picked and not others? For example, why IIα and not IIβ?.

2. In line 68, beome instead of become

3. In line 274, the authors suggest that the C-terminal alleles affect the biofilm formation based on figure 2C. A similarly strong statement is made in line 312. In figure 2C, apart from varying the C-terminal allele, the authors are also changing the N-terminal alleles. I would strongly recommend the authors to test this theory by comparing biofilm formation of cells harboring the same N-terminal but different C-terminal alleles, for example Iα and Iβ.

4. In the section starting in line 447, the authors compare the distribution of OmpA alleles in other gram-negative bacteria. They use sequences of different E. coli pathotypes, however they do not included any AIEC (adhesive-invasive Escherichia coli) strains. It is important to note the genetic similarities between AIEC and ExPEC, despite its niche. A study of the OmpA variants found in AIEC (and some ExPEC) has previously published 10.3389/fmicb.2019.01707. I would strongly recommend the authors to take a look at this paper and the OmpA variants that they describe.

Reviewer #2: Overall, the experiments in this study are well described and clearly presented. Strengths and weaknesses of the data are addressed appropriately, and the results provide new and interesting insight into the phenotypic effects of OmpA variants. There are a few key controls and other mostly small issues that should be addressed to strengthen the paper:

- To better understand the results, it would help to know if the OmpA allelic variants are expressed at similar levels, have similar turnover rates, or if they reach the outer membrane at similar levels. This could at least be noted as a potential confounding issue for interpretation of the results.

- Use of an empty vector control in MG1655∆ompA would be a better comparator for the OmpA allelic variant expression strains in the phenotypic assays.

- Abstract could be revised to make the point of the paper and its findings more apparent to a broader audience.

- Authors often use the word “terminal” as a noun when referring to the N- or C- terminus or terminal domain.

- Shigella is basically E. coli. The authors’ argument that ompA might be horizontally transferred between these two types of bacteria was not clear.

6. PLOS authors have the option to publish the peer review history of their article (what does this mean?). If published, this will include your full peer review and any attached files.

Reviewer #1: No

Reviewer #2: No

---

## [Author Response · Author response to Decision Letter 0]

5 Sep 2022

"Funding for this work was provided by the National Science Foundation grants CBET-1236510 and CBET-1604576, and the Unites States Department of Agriculture National Institute of Food and Agriculture, award number 2017-6702-26137. The funders had no role in study design, data collection and interpretation, or the decision to submit the work for publication."

"Funding for this work was provided by the National Science Foundation (https://www.nsf.gov/) grants CBET-1236510 (MLS) and CBET-1604576 (LRJ), and the Unites States Department of Agriculture National Institute of Food and Agriculture (https://www.nifa.usda.gov/), award number 2017-6702-26137 (LRJ). The funders had no role in study design, data collection and analysis, decision to publish, or preparation of the manuscript."

The funding statement has been removed from the manuscript. No edits are needed for the acknowledgement section.

 

Reviewers' comments:

Reviewer's Responses to Questions

Comments to the Author

We thank the reviewers for their time spent reading this work and providing their constructive feedback.

1. The introduction should also give a small description of ExPEC, and what other groups form them (APEC, UPEC, NMEC). Those are mentioned during the manuscript, but a short description or definition at the introduction would help the reader. 

We have edited the text so that all ExPEC types are defined in the introduction.

2. Also, why were those 7 alleles picked and not others? For example, why IIα and not IIβ?.

This is a choice that was made early in the project and that we have since wished that we could redo. These alleles were selected and tested before we recognized the breadth of the allelic variation, and by the time that realization was made, the graduate student who performed the various characterizations had already left for a postdoctoral researcher position, and testing of any other alleles was outside the scope of our other funding. We are hoping that this publication will help us acquire a fresh round of funding so that we can investigate these alleles further.

2. In line 68, beome instead of become

This has been corrected – thank you

3. In line 274, the authors suggest that the C-terminal alleles affect the biofilm formation based on figure 2C. A similarly strong statement is made in line 312. In figure 2C, apart from varying the C-terminal allele, the authors are also changing the N-terminal alleles. I would strongly recommend the authors to test this theory by comparing biofilm formation of cells harboring the same N-terminal but different C-terminal alleles, for example Iα and Iβ.

This reviewer has made a suggestion that we strongly agree with. For the same reason described above, we have not been able to do these experiments. It is our hope that this manuscript, if published, can be used as the basis for funding for a new project, so that we can perform this type of experiment (and many others).

4. In the section starting in line 447, the authors compare the distribution of OmpA alleles in other gram-negative bacteria. They use sequences of different E. coli pathotypes, however they do not included any AIEC (adhesive-invasive Escherichia coli) strains. It is important to note the genetic similarities between AIEC and ExPEC, despite its niche. A study of the OmpA variants found in AIEC (and some ExPEC) has previously published 10.3389/fmicb.2019.01707. I would strongly recommend the authors to take a look at this paper and the OmpA variants that they describe.

We are very grateful to the reviewer for making us aware of this publication. There are so very many publications about OmpA, it is difficult to keep track of them all. Thanks to the reviewer’s suggestion, we have used the referenced findings to update Figure 5, Table 2, and Figure S1 (A and B). Specifically, we found that the (III, �) allele is enriched in AIEC relative to APEC isolates, just as this same allele is enriched in UPEC relative to APEC isolates.

Reviewer #2: Overall, the experiments in this study are well described and clearly presented. Strengths and weaknesses of the data are addressed appropriately, and the results provide new and interesting insight into the phenotypic effects of OmpA variants. There are a few key controls and other mostly small issues that should be addressed to strengthen the paper:

- To better understand the results, it would help to know if the OmpA allelic variants are expressed at similar levels, have similar turnover rates, or if they reach the outer membrane at similar levels. This could at least be noted as a potential confounding issue for interpretation of the results.

Done.

- Use of an empty vector control in MG1655∆ompA would be a better comparator for the OmpA allelic variant expression strains in the phenotypic assays.

We agree that these experiments are appealing. However, deletion of ompA leads to poor cell physiology. This observation was reported by, for example (Wang Biochem Biophys Res Commun 2002), and also observed in our attempts to use an ompA deletion mutant as a control in (Liao et al 2017).

- Abstract could be revised to make the point of the paper and its findings more apparent to a broader audience.

Done.

- Authors often use the word “terminal” as a noun when referring to the N- or C- terminus or terminal domain.

Thank you for pointing this out. We have corrected this throughout the manuscript.

- Shigella is basically E. coli. The authors’ argument that ompA might be horizontally transferred between these two types of bacteria was not clear.

We have edited the text of the results section and the abstract in response to this comment.

---

## [Decision Letter · Decision Letter 1]

29 Sep 2022

Allelic Variation of Escherichia coli Outer Membrane Protein A: Impact on Cell Surface Properties, Stress Tolerance and Allele Distribution

PONE-D-22-18140R1

Dear Dr. Jarboe,

We’re pleased to inform you that your manuscript has been judged scientifically suitable for publication and will be formally accepted for publication once it meets all outstanding technical requirements.

Kind regards,

Olivia Steele-Mortimer, Ph.D.

Academic Editor

PLOS ONE

Additional Editor Comments (optional):

Reviewers' comments:

Reviewer's Responses to Questions

**Comments to the Author**

1. If the authors have adequately addressed your comments raised in a previous round of review and you feel that this manuscript is now acceptable for publication, you may indicate that here to bypass the “Comments to the Author” section, enter your conflict of interest statement in the “Confidential to Editor” section, and submit your "Accept" recommendation.

Reviewer #1: All comments have been addressed

Reviewer #2: (No Response)

2. Is the manuscript technically sound, and do the data support the conclusions?

Reviewer #1: Yes

Reviewer #2: Yes

3. Has the statistical analysis been performed appropriately and rigorously? 

Reviewer #1: Yes

Reviewer #2: Yes

4. Have the authors made all data underlying the findings in their manuscript fully available?

Reviewer #1: Yes

Reviewer #2: Yes

5. Is the manuscript presented in an intelligible fashion and written in standard English?

Reviewer #1: Yes

Reviewer #2: Yes

6. Review Comments to the Author

Reviewer #1: The authors have improved their manuscript and addressed all my previous comments. I will recommend the authors to give it a last look for typos before publishing it. For example, there is a ( missing in line 111.

Reviewer #2: The authors addressed by concerns adequately

7. PLOS authors have the option to publish the peer review history of their article (what does this mean?). If published, this will include your full peer review and any attached files.

Reviewer #1: No

Reviewer #2: No

---

## [Editor Report · Acceptance letter]

4 Oct 2022

PONE-D-22-18140R1 

Allelic Variation of *Escherichia coli* Outer Membrane Protein A: Impact on Cell Surface Properties, Stress Tolerance and Allele Distribution 

Dear Dr. Jarboe:

I'm pleased to inform you that your manuscript has been deemed suitable for publication in PLOS ONE. Congratulations! Your manuscript is now with our production department. 

Kind regards, 

on behalf of

Dr. Olivia Steele-Mortimer 

Academic Editor

PLOS ONE